*"Reborn in Guate":*
*Making Resource Frontiers in Asylum in Guatemala's Northern Petén*

On a sunny June afternoon in 2022, the cathedral square above the pastel-painted town of Isla de Flores is abuzz with activity. Banners are festooned across a far corner of the square, publicising the celebration of World Refugee Day in the Petén: Guatemala's northernmost region. Under the slogan 'Reborn in Guate,' the United Nations High Commissioner for Refugees' (UNHCR) newly established Petén office has put together a programme of art, dance, music, and poetry.[1] Knitted clothing and homemade sweets are for sale on long tables, while a gallery of artwork depicting narratives of resilience is displayed in erected white canvas tents. A raised stage at the centre of the tents provides a platform for a succession of folkloric acts throughout the day. Guatemala is well-known as a country with tragic histories of conflict. Hundreds of thousands of Guatemalans left the country in the face of US-instigated civil war violence and genocidal policies against Indigenous Maya (Schirmer 1998).[2] However, these celebrations are not for refugees from Guatemala, but rather migrants claiming asylum from surrounding regions *to* Guatemala. A young Honduran teenager stands on the stage, rapping about the hardship of long journeys. He is followed by an older Honduran man, who gives a speech emphasising the kindness shown by people locally. A canvas tent on the side is open for visitors, providing information about the resettlement support offered by local organisations.

In Flores and Santa Elena, the regional capital of the Petén, located across a causeway from the island town, the concept of claiming asylum to – not from – was little heard of prior to the last decade. Now, public information campaigns on asylum in Flores – one of many new 'Cities of Solidarity' around Guatemala – are inescapable.[3] Not far from the Refugee Day celebrations, tourist-style posters in Santa Elena's central bus terminal extoll Guatemala as an asylum destination. One stretches dramatically along the façade of the arrivals hall, so as to be strategically visible to those arriving at the station. The poster is fringed with stick figures of people running for safety and the logos of UNHCR and the Guatemalan non-governmental organization, *El Refugio de la Niñez*.[4] At its centre, it features a family holding hands as they clamber over train tracks. In large blue font, with the words 'danger,' 'protection,' and 'refugee' highlighted, it reads:
> "If your life is in danger, and you cannot return to your country, you can ask for protection as a refugee in Guatemala. We can help you!" (*Si tu vida corre peligro, y no puedes regresar a tu pais, puedes pedir proteccion como refugiado en Guatemala. Te podemos ayudar!*).

Around another corner in the terminal, signs point towards a small office, the Attention Center for Migrants and Refugees. There, seven institutions converge in what UNHCR staff locally term their 'one-stop shop' – the *Procurador de los Derechos Humanos* (Human Rights Defenders),

---

[1] UNHCR is known by the acronym ACNUR in Spanish-speaking countries. However, in this article I use their English acronym UNHCR as it is globally recognisable.

[2] In Guatemala, the term Maya refers to over 22 diverse sociolinguistic groups, who collectively comprise approximately 60% of the population. There are also other Maya groups across Meso-America including in neighbouring Mexico, El Salvador, and Honduras. Although 'Maya' is a homogenising umbrella term that fails to capture the cultural nuances of different groups, identifying as Maya is a way to resist neo-colonial erasure, reaffirm one's heritage and expose the arbitrariness of international border-making. See Lopez Casertano (2022) for an excellent discussion into Mayan experiences in contemporary Guatemala or *Iximulew* ('land of corn').

[3] A number of cities across Mexico, Central and South America have elected to take part in UNHCR's Cities of Solidarity initiative, including nine cities/departments in Guatemala. See Morris (2021a) for a discussion into marketing methodologies used by UNHCR to 'sell' refugees to new audiences, even as the focus on 'refugees not migrants' habitually leads to the exclusion of many people.

[4] *El Refugio de la Niñez* ('The Children's Shelter') is a Guatemalan NGO set up in 2009 to focus on the rights of children and adolescents in situations of violence, risk, and vulnerability. Since this time, they have partnered with UNHCR in a number of programmes centered on asylum seeker and refugee populations.

*Conamigua* (the National Council for Attention to Migrants of Guatemala), *Medicos Sin Fronteras* (Doctors Without Borders), *Cruz Roja* (Red Cross), UNHCR, and *El Refugio de la Niñez.* Inside*, Guatemalan social workers wait patiently for anyone wanting to learn more about claiming asylum in Guatemala. UNHCR information signposts advising these services are also strategically located along the highway that cuts through the Petén: sandwiched between Honduran and Mexican borders and passed through by migrants making long journeys northward. Just a few blocks down from the bus terminal is UNHCR's new regional office. On occasion, their fleet of distinctive mobile unit vans are visible, having returned from advertising claiming asylum in Guatemala along the country's major migration routes and border posts with Mexico.[5]

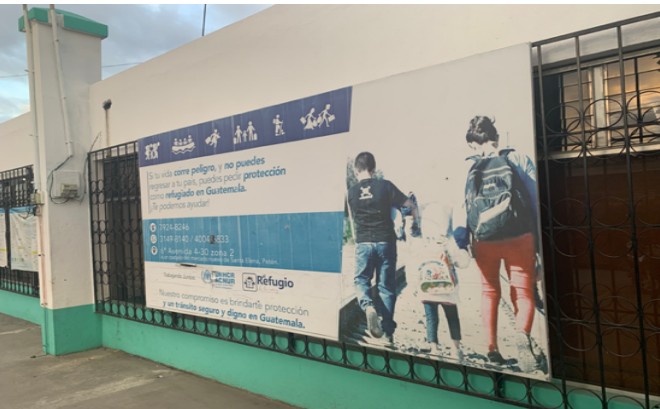

**Figure 1:** Advertising asylum for Guatemala at Santa Elena's bus station, photograph by author.

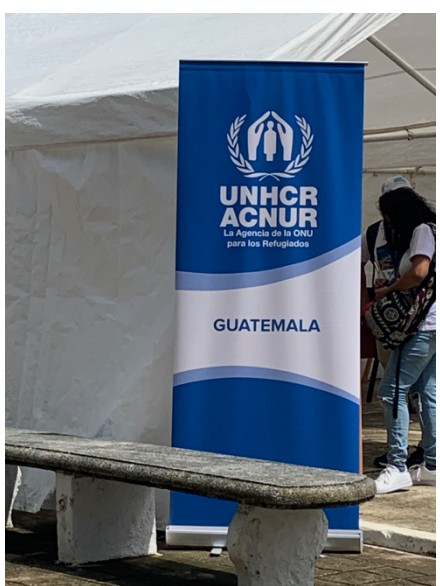

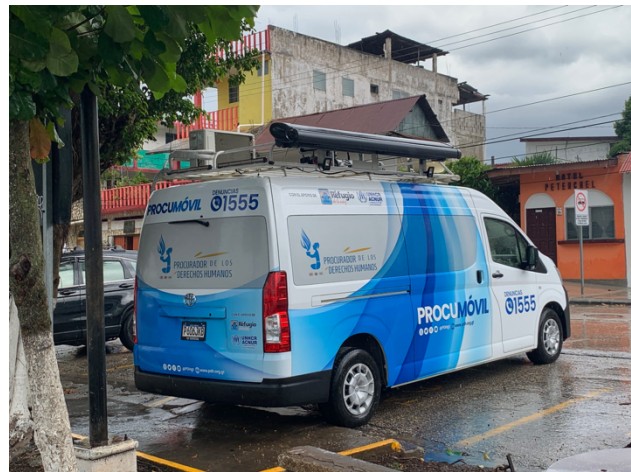

**Figure 3:** UNHCR mobile units for advertising asylum in rural regions, photograph by author.

**Figure 2:** World Refugee Day Petén, photograph by author.

What explains these dramatic transformations taking place in Guatemala's rainforested Petén region? In the last five years, Guatemala's national asylum and resettlement system has received a significant boost of support from the US government. As part of a regional approach begun in 2017, known as the Regional Comprehensive Protection and Solutions Framework (or its Spanish

---

[5] UNHCR and *El Refugio de la Niñez* also have a mobile boat unit, 'Proculancha,' in Lake Izabal, near the border with Honduras, which performs a similar function. They also launched a new WhatsApp chatbot that provides information on Guatemala's asylum system.

acronym MIRPS), Guatemala, together with Mexico and other Central American countries (Belize, Costa Rica, El Salvador, Honduras, and Panama), is steadily developing the capacity to receive and support asylum claims in the country. This strategy differs from extraterritorialised asylum, where destination country authorities conduct asylum procedures and provide housing and some humanitarian support (Endres de Oliveira and Feith Tan 2023). Instead, it involves funding the entire development of national refugee processing and resettlement practices.[6]

This article advances a new theoretical framework to examine the surge in new asylum regimes worldwide, and specifically in Guatemala. I look at these recent developments through the lens of 'resource frontiers' to emphasise how political, economic, and moral capital is extracted from migrants as part of an expanding extractive frontier. As a theoretical concept, the resource frontier is used to explain recent frontier-makings around the extraction of resources (Cons and Eilenberg 2019; Tsing 1993). Remote areas are reconfigured as 'zones of opportunity,' involving new articulations of territorial governance, as well as regional and international networks of accumulation and security. Studies of resource frontiers have focused on more typically thought-of extractive spaces, such as where new mineral or petrochemical resources have been discovered (Watts 2018) or monocultural crop booms (De Koninck et al. 2011). I see *asylum frontiers* as the social spaces connected to the exploration and development of a resource sector that centres on extracting value from people on the move. This framing allows for more attuned scholarship to the extraction of value as a driving force for expanding asylum, provoking important conversations into alternative migration pathways beyond asylum. In studying these developments, I argue that it is precisely this value economy around asylum which advances a frontier marked by precariousness and uncertainty. Underneath Guatemala's promotional spectacle, few migrants are interested in claiming asylum locally. Meanwhile, there is little support for returnees or internally displaced persons in Guatemala. These concerns raise questions as to whether refuge can be found when harsh urban conditions are faced by Guatemala's own citizens across the country. Such aspects are conspicuously absent from the promotional fervour around asylum in Guatemala. Instead, the imaginary of Guatemala's new asylum regime provides an assortment of politicians, NGO, corporate, and state contractors with opportunities to reap the benefits of a highly visible intervention.

In making these arguments, I build on my ethnographic fieldwork in three very different places – Guatemala, Jordan, and Nauru – to trace the emergence of new asylum frontiers. Over the last near decade, I have been tracking new fronts of extraction related to expanding asylum regimes. I conducted ethnographic fieldwork in Australia, Geneva, Fiji, and Nauru for a total of fifteen months between 2015-2016; in Jordan for three months across 2018-19; and in Guatemala across 2021-2023. The case of Nauru was particularly extreme in that the island nation's entire economy soon revolved around the asylum industry (Morris 2019, 2021b, 2023a). Overlapping regimes of resource governance buoyed mineral and migrant processing regimes as asylum replaced phosphate as a new but equally resource cursed extractive sector. Local tensions, self-harm, and other forms of extreme exploitation characteristic of capital-intensive resource extraction zones coursed through Nauru's new national asylum regime.

---

[6] Some Central American countries, in particular Costa Rica, already have long-established traditions as places of asylum for migrants leaving repressive regimes (Basok 1993; Hammoud-Gallego and Freier 2022). These practices have become more institutionalised and widespread with the convergence of US policies targeting the securitisation of migration.

In Jordan, I witnessed the burgeoning growth of the asylum industry as an array of NGOs and UN agencies received contracts connected to Syrian refugees (Morris 2020). International NGOs, such as the International Rescue Committee and the Danish and Norwegian Refugee Councils, brought immense material resources that were inaccessible to refugees from other regions (namely Somalia, Sudan, and Yemen). Meanwhile, the Jordanian government used refugee rights as a strategic instrument of foreign policy: what Victoria Kelberer (2017) and others have termed 'refugee rentierism' or 'refugee rent-seeking' strategies (Freier et al. 2021). But I found that the motivations of the Jordanian government and contracted workforces could not be reduced to economic interests alone. Many individuals were sympathetic to the experiences of migrants displaced from nearby regions. This logic of moral value is set within Jordan's history as a host state for substantial numbers of Palestinian and Iraqi refugees.

Since then, I have continued to follow the extraction of value through to Guatemala. I conducted fieldwork in Guatemala City and Guatemala's northern Petén region over three separate month-long field trips in 2021, 2022, and 2023. During this time, I interviewed or spoke informally with asylum seekers and refugees for Guatemala, representatives from relevant government departments, UNHCR, and NGOs involved in Guatemala's new asylum system. In 2023, 597 asylum applications have been filed to the country's new National Commission for Refugees (*La Comisión Nacional para Refugiados* or CONARE).[7] At the time of writing, 872 refugee visa holders and 2,348 asylum-seekers resided in the country.[8] The Petén had the highest number of claims after Guatemala City (a 10% increase from that same period in 2021), but still incredibly low numbers compared to other global regions. In addition, the number of Guatemalans returnees almost doubled overall from 40,650 in the first nine months of 2021 to 73,610 in the same period in 2023 (IOM 2023a): many of whom experience little support, while all-too-often coping with financial and emotional strains. I attended high-level migration meetings and took part in multi-day rangers' trips to the northern Guatemala-Mexico border: one of many employment schemes for foreign nationals on refugee visas in Guatemala. I also spoke informally with migrants making journeys northward through the Petén, many of whom stay at the network of *Centro Migrantes* that provide accommodation and food for stays up to five of days.

My first research period coincided with Kamala Harris' first international trip abroad as Vice President in June 2021. In a controversial speech at the *Palacio Nacional de la Cultura* in Guatemala City, Harris pledged a tough on borders approach. As part of their so-called Root Causes Strategy, Harris is heading the Biden Administration's efforts to advance local livelihoods in the Northern Triangle region of Central America: El Salvador, Guatemala, and Honduras. These efforts include funding a range of employment opportunities, as well as pushing for the advancement of regional asylum systems to dissuade migrants' asylum claims to the US. Many of the organisations I met were familiar faces, as were the forms of cultural production, such as World Refugee Day, designed to naturalise commonplace assumptions of people as refugees. Encountering these forms of mobile labour and expertise pushed me to understand the externalisation of asylum as a 'frontier assemblage' (Cons and Eilenberg 2019). Material and discursive, human and non-human agencies are all involved in shaping the configurations of emerging asylum frontiers. Certainly, the diverse geographical contexts of Guatemala, Jordan, and

---

[7] CONARE is an inter-ministerial body created as an advisory body to the National Migration Authority (*Autoridad Migratoria Nacional* or AMN), which is the institution responsible for Guatemala's migration policies.
[8] See UNHCR's new operational data portal on Guatemala: https://data.unhcr.org/en/country/gtm

Nauru have their own particularities: Guatemala is itself a settler colony based on *criollo* notions of nation building and governed by a *ladino* elite (Martínez Peláez 2009).[9] By bringing these sites together, I gloss over many social, political, and economic specificities. But in so doing, my intention is to expansively consider the global trend towards offshored and outsourced asylum models as they zig-zag from north to south.

In what follows, I ground my theoretical framework of asylum frontiers in Guatemala's northern Petén region. This first section considers the purchase of the resource frontier analytic in making visible contemporary asylum frontiers and their histories. Thinking about asylum externalisation as resource frontiering also de-exceptionalises geographies of containment and control. The Petén is not unique; rather, it illustrates the forms of exploitation and administrative violence baked into the racialised mobility regime of the international refugee system. Non-western and largely people of colour are subject to an administrative regime that demands they prove suffering to move elsewhere. The third and fourth sections take their inspiration from Dolly Kikon's (2019) work on oil and coal mining to trace how different political actors – migrants, Indigenous Mayan refugees and deported Guatemalans – 'live with' these frontier economies. These sections ask how the development of a national asylum system in Guatemala sits with the country's own histories of asylum and enforced return. The article closes by revisiting the consequences of these new insitutionalised asylum locations. Given that state and non-state agencies are continuously investing in asylum expansion, the focus on asylum frontiers contributes substantially to the interrogation of asylum. This analysis is necessary to make sense of the realities facing migrants today. By effectively critiquing the promotion of asylum, we might better consider alternatives that support local livelihoods, rather than driving the perpetual expansion of asylum frontiers.

**Section I: Asylum Frontiers**
The rapid transformation of agrarian and forest spaces into sites of intense resource extraction have galvanised an interest in the study of resource frontiers (Cons and Eilenberg 2019; Kikon 2019; Tappe and Rowedder 2023; Tsing 2008). As a theoretical concept, the resource frontier is used to explain recent frontier-makings around the extraction of resources. Resource frontiers are often taken to be the incorporation of marginal spaces and ecologies as coveted zones of potential. They are highly imaginative in that these are spaces where the material realities of place are interwoven with different visions and cultural vocabularies. Socially constructed representations of resource frontiers vary. They are at once construed as crucial spaces for economic activity (Patel and Moore 2017), dangerous lawless sites outside of the optics of state control (Tsing 2005), and an untouched wilderness ripe with possibilities (Li 2014). These often-fictive framings of frontiers all centre in some way on the critical need for intervention. Research in these fields has long considered natural resource extraction and the rapid transformation of remotes spaces into socially and ecologically destructive production sites.

In recent years, more marginal spaces are being transformed into locations for refugee processing and resettlement across the Global South. These policies produce similarly transformative effects to those described by political ecologists in the production of resource frontiers. Extractive processes push many people to move from homeland regions. This includes massive land-grabs, ecological destructions from mining, and the implementation of neocolonial development

---

[9] In Guatemala, the term *ladino* signifies a mix of European and Indigenous Maya ancestry that attempts to suppress Indigenous backgrounds (Montejo 2005).

programs, all of which generate migrations, new resources and profits (Sassen 2014). People are then subject to the extractive systems of legal technologies that track their movements across territories (Walters 2004). Here, the resource consists of racialised migrants, whereby the bureaucratic practice of refugee classification serves to regulate people's movements and value (Morris 2021a). Migrants are made valuable to humanitarian organisations and state agencies because of their designation as vulnerable people (Coddington et al. 2020; Martin and Tazzioli 2023). But this is not to imply that migrants are outside the asylum industry in terms of the biopolitical function of their labour power. Rather, as I have argued elsewhere, asylum claimants are also entangled in the 'intimate labour' of making an asylum claim (Morris 2023c).

For people on the move in Guatemala, humanitarian aid and international protection forces them to modify their narratives. The compounded reasons for migrating – criminal violence, poverty, natural disasters, and environmental deterioration – are not causes in the Geneva Refugee Convention framework, upon which refugee certification is decided. Migrants are pushed to identify as – and perform – what Liisa Malkki (1995) has referred to as 'refugeeness' in order to legalise their movement across borders. This "embodied performance of trauma" (Pine, 2020: 212) entails recounting intimate experiences and narratives of trauma in order to move elsewhere. Asylum policies then change physical spaces and territories in line with new forms of human extraction, as I experienced ricocheting from the Pacific through to the Middle East and Central America. Like the resource extractive impacts typically described by political ecologists (Watts and Peluso 2013), these epicentres of extraction and production often wreak havoc on migrants and local communities (Morris 2019). As this article goes on to show, the forms of violence produced by asylum frontiers are also site-specific phenomenon rooted in local histories and social dynamics, yet connected to macro processes of material transformation and power relations.

The development of an asylum regime in Guatemala is indicative of a shift in border relations. Most forced migrants are hosted in countries in the Global South, proximate to conflict or other disaster zones. However, the last 15 years have seen a dramatic increase in Western countries funding the asylum procedures of countries in the Global South (Dastyari et al. 2022; Morris 2023a). The EU has adopted this model in Eastern Europe, Turkey, North Africa, and Central Asia (Follis 2012; Gazzotti 2021; Lambert 2021), as has the UK in their recent attempted arrangement with Rwanda (Morris 2023b). Australia uses migrants as a form of economic development with Pacific island nations and across south Asia (Morris 2023a; Nethery and Gordon 2014). The US has long utilised military bases for migrant interdiction efforts, as well as funding detention centres in other regions and working with law enforcement officials from nearby countries to limit migrant routes (Loyd and Mountz 2018).

This trend of developing national asylum systems has found its way to Guatemala. Following pressure and financing from the US, Mexico developed a southern border program in June 2014, known as *Programa Frontera Sur*, to strengthen migration control measures along the Mexico-Guatemala border. Previously, the majority of those making these challenging journeys traveled through the south-western Guatemalan states of San Marcos and Huehuetenango. Heightened US-funded border enforcement in these regions has pushed more people to pass through the northern Petén border, where I conducted my research. The Petén is also the main 'transit region' for those making their way north from Honduras. Most Honduran and Salvadoran migrants intending to travel to the US, but unable to meet the strict visa protocols, will pass through Guatemala. In recent

years, more migrants from the Caribbean, South America, Africa, and Asia have sought to reach the US through the Central American isthmus (Selee et al. 2023). Without the ability to easily regularise their movement, many turn to the international refugee regime. They now encounter the buffer zones that unfold as new types of resource governance emerge through outsourced asylum regimes.

There is now a powerful body of literature on border externalisation (Bialasiewicz, 2012; Boswell 2003; Freier et al. 2021; Gazzotti et al. 2023; Vammen et al. 2022). This includes such strategies as maritime and third-country interceptions, biometrics, carrier sanctions, agreements and border control training with so-called 'transit' countries, public messaging campaigns, and extraterritorial claims processing and detention. Some of the scholarship flagging up these trends focuses directly on outsourcing asylum (Dastyari et al. 2022; FitzGerald 2021; Hyndman and Mountz 2008; Moreno-Lax 2017; Morris 2023a). Following this model, countries in the Global North outsource asylum processing and refugee resettlement to neighbouring states in exchange for financial and development support. Alongside the research that visibilises these moves, there is also important work centred on the migration industry: a vast industry of corporate, non-governmental, government, solidarity campaigners, and other actors characterised by profit-making activity from people on the move (Andersson 2022; Cranston et al. 2018; Franck 2018; Gammeltoft-Hansen and Sørensen 2013; Golash-Boza 2009; McGuirk and Pine 2022; Morris 2017, 2021a). More specifically, contracted workforces, politicians, and media workers gain through the representation of border securitisation. Electoral support for right-wing anti-immigrant platforms rides on the production of hysteria around the Other. From border security and detention management corporations to healthcare providers, humanitarian organisations, legal firms, and research institutes, a range of actors have a stake in the migration industry, and in increased securitisation. Asylum is not only transferred to the private sector but also southern state governments, whereby economic logics of profit run through dispersal (Freier et al. 2021).

My work draws on these two bodies of scholarship to develop the theoretical framework of 'asylum frontiers.' This analytic looks to capture the transformation of seemingly marginal spaces into locations for refugee processing and resettlement. Here, I offer three rejoinders to the conceptual terrain of the migration industry and migration externalisation literature. First, I suggest that rather than focusing on industry actors, what matters is the assemblages of materialities, cultural logics, political economic processes, ecologies, and actors that facilitate and condition mobility (Cons and Eilenberg 2019; Morris 2021b; Xiang and Lindquist 2014). To think of migration governance in this Deleuzian sense enables us to see the social and material elements that rework remote spaces into new kinds of productive sites – sites slated for refugee processing and resettlement. Second, I emphasise a focus on *value* over profit as a way of considering the many motivations of actors involved in advancing new asylum frontiers. Value generation takes numerous forms, including political, economic, symbolic, as well as moral value. Committed humanitarian but also corporate and state workforces are not just on the hunt for investment opportunities. While wanting to eke out a living, many are trying to better people's lives (Malkki 2015). At the same time, asylum also holds use-value for migrants, who are entangled in an imbalanced performative economy. Migrants, particularly from the Global South, must produce 'pictures' of lives eligible for protection that fit templates of victimhood and vulnerability (Cabot 2013). In this way, many migrants work to transform themselves into an economic asset through

refugee status (Bardelli 2020): a form of unfree labour "that is at once exploitative and generative of new forms of belonging" (Calvão 2016: 456).

This leads to my third contribution in that much of the scholarship on the migration industry and migration externalisation tends to reinforce the international refugee regime even as they critique it. States that take on third country asylum arrangements are decried as rife with human rights abuses or not holding the capacity to take on such modernist technical endeavours. In developing the asylum frontiers analytic, I begin with the premise advanced by some critical migration and refugee studies researchers and migrants that asylum is structurally bound up in the governance of race and place (Bhagat 2024; Espiritu 2014; Mayblin 2017; Picozza 2022). From its colonial foundations, asylum has functioned by creating categories and ideologies that legitimate various capitalist modes of accumulation and nation state projects. To this day, the international refugee regime operates as a form of racialised border control, fostering life hierarchies and reproducing racial categories in the present (Pallister-Wilkins 2022). Representations of Global South violence marshalled as anti-externalisation advocacy are dehistoricising and demeaning, reeking of white western salvationalism (Danewid 2017; Morris 2023a). Such Conradian imaginaries also provide more moral capital and mobile labour to the overall asylum system that supports the expansion of asylum frontiers.

Guatemala's asylum frontier has arisen in relation to conditions of what Feldman et al. (2011) term 'the accumulation of insecurity.' Successive American governments and many news media outlets have accrued political economic value through the production of hysteria around the Other (Bigo 2012). Some American politicians and news media outlets regularly perpetuate notions that Latinos, in particular, are "an invading force" bent on "destroying the American way of life" (Chavez 2013: 3). Blame for the country's deep-seated inequalities is habitually placed on racialised migrants through discourses of 'welfare scrounging' and 'job theft.' But rather than simply bulking up the enforcement structures of their immediate borders and those of surrounding states, the US – like many western states – simultaneously utilises the international refugee system. Governments have non-derogable duties under international human rights, refugee, humanitarian and customary law. Article 33 of the Convention, known as the *non-refoulement* clause, asserts that refugees must not be returned where they face serious threats to their life or freedom. States reason that, provided they do not directly contravene this prohibition, they can send refugees to other countries. Through coordinated partnerships around asylum, states look to outsource, but not completely derogate on, their international protection responsibilities. However, as researchers point out, this kind of arrangement flagrantly disregards commitments to international refugee law (Dastyari et al. 2022). And in the case of countries such as Guatemala, where support for its own citizens is meagre, US deflection of asylum to Guatemala might arguably constitute *refoulement*.

The use of the refugee regime in this way is fitting. Although masked by humanitarian precepts, this Eurocentric system was actually set up as a gatekeeping apparatus to control mobility and labour (Behrman 2019). For Refugee Convention signatories, funding the asylum procedures of countries in the Global South is seen as a form of 'burden sharing' (Boswell 2003). According to this argument, the growth in southern asylum systems will encourage more states to play a part in supporting refugees locally, and ultimately restrict migrants from claiming asylum at US borders. It enables government agencies to refute the charges of refugee solidarity groups that migrants cannot seek asylum in transit countries because of a lack of a functioning asylum system. By

advancing third country asylum policies, the US theatrically displays nation-state sovereignty together with an alleged first world humanitarianism. This Janus-faced approach aims to fulfill the double role of 'saving lives' and 'combating illegal immigration,' engendering a form of humanitarian borderwork that simultaneously cares and controls (Pallister-Wilkins 2022). Indeed, in Guatemala, the promotion of local narratives of refugee protection exists alongside fear-based messaging campaigns, also funded by the US, that highlight the risks of migrant journeys northward (Morris 2022). The US also pressures the Mexican government to stop migrants at their border with Guatemala, and Guatemala at its border with Honduras. This contrasting performance of local protection and deterrence reveals how asylum in Guatemala is a smoke screen to conceal the aim of keeping migrants away from the US border.

For the Guatemalan government, more refugees in Guatemala means more American financial investment. In the last fiscal year, bilateral, regional, and humanitarian assistance through the Department of State and USAID averaged $231.3 million per year.[10] The sort of humanitarian pageantry that I encountered in the Petén also enables Guatemalan politicians to position the country as a haven of democracy, liberty, and universal rights. Yet, underneath the display of Guatemalan refugee protection, few migrants are interested in claiming asylum locally. Most migrants passing through the Petén end up applying for asylum in Mexico or head to the US. In Mexico in 2023, asylum applications topped 141,053 claimants: many of whom are also Guatemalan.[11] Significant numbers of migrants are also detained and eventually deported from Mexico: far more than from the US. Between 2019 and 2023, it is estimated that Mexico deported around 500,000 people, while the number of migrant detentions in Mexico was almost 700,000 in 2023 (Alba 2024). That the numbers of migrants who actually claim asylum to Guatemala is very low – and that Guatemalans are still leaving the country in substantial numbers – speaks to the symbolic value of the humanitarian-securitisation spectacle. Such a scarcity of migrant resources is largely immaterial to the extraction of value. Rather, the performance of asylum in Guatemala facilitates larger transnational flows of capital. American politicians, NGOs, corporate, and state contractors extract value from a highly visible spectacle of border enforcement. This "economy of appearances" (Tsing 2000), where dramatic performance is key to spectacular accumulation, is a regular feature of the hunt for capital in frontier spaces.

The buildup of Guatemala's asylum capacity came just prior to the US' high-profile asylum and transit ban. Under the 'asylum ban' policy, announced in May 2023, large-scale restrictions are imposed on who can apply for asylum at the US southern border. The majority of those who travel through Mexico and Central American countries must seek asylum there first.[12] Then in June 2024, not long before the presidential elections, the Biden administration augmented this further. They issued an executive order that prevents migrants from lodging asylum claims at the US-Mexico border when undocumented border crossings surpass a threshold of 2,500 people. The US government is scaling back domestic asylum procedures but funding its development elsewhere. By financing Guatemala's asylum system and that of surrounding countries,

---

[10] See https://www.foreignassistance.gov/cd/guatemala

[11] See https://www.gob.mx/comar

[12] This is unless an asylum seeker can secure a limited appointment through the CBP One phone application: part of what Lupe Flores (forthcoming) describes as a 'digital externalization' of the Mexico-US boundary. Critics charge that appointments are scarce, the phone application is flawed, and some migrants do not have access to the smart phones required for downloading the application (Verduzco 2023). In June 2023, the US government also launched the Safe Mobility Offices initiative with UNHCR and IOM. Under this programme, Guatemalan nationals wanting to travel to the US under various migration pathways, such as the H-2A temporary agricultural work visa, await decisions in Guatemala. The goal of all these initiatives is to reduce the numbers of migrants at the Mexico-US border.

American policymakers are utilising asylum as a tactic of enforcement to decrease migration to the US. In this way, Guatemala has once again become an exploitable frontier zone, shaped by various actors' interests.

The emergence of asylum frontiers depends on an extensive assemblage of state and non-state actors, including international organisations, governments, private companies, NGOs, and academic research institutes (Cabot 2016; Franck 2018; Morris 2021a). In fact, the asylum industry, as I and others term it (McGuirk and Pine 2022; Morris 2019), is nothing short of gargantuan. Many of the familiar organisations I observed in Nauru and Jordan, such as the UNHCR, the Red Cross, Doctors Without Borders, and the International Organization for Migration, expanded their operations to the Petén. These organisational leviathans, like the UNHCR, whose total expenditure in 2023 alone was $5.167 billion, grew to have a substantial presence locally. But alongside clear fiscal interests, many of these workforces come with humanitarian motivations. The imaginary of the refugee regime as a benevolent system that supports migrants in need is part of the 'affective economies' (Ahmed 2004) of value extraction and generation that drives many people in this field. This abstract imaginary forms a crucial part of the assemblage that unfolds in new asylum frontiers.

The development of local refugee legal systems, bureaucracies, and resettlement services through to forms of cultural production, such as World Refugee Day Petén and Cities #WithRefugees Campaigns, all contribute to fostering asylum in new localities. In the Petén, international organisations, whose work centres on refugees, combine with newly developed Guatemalan organisations, including *El Refugio de la Niñez*, and state migration agencies to promote asylum locally and then provide various forms of material resources. In addition, the Guatemalan government set up a series of national institutions and laws in order to legally 'produce' more refugees locally, and so acquire US financial support. In 2019, Guatemala's National Migration Authority issued a 'Regulation on the Procedure for the Protection, Determination and Recognition of the Status of Refugees.' This regulation created the National Commission for Refugees (CONARE) as an interministerial advisory body to support the National Migration Authority in adjudicating migrants' asylum claims. Staffed with lawyers, social workers, and psychologists, the aim of this body is to adjudicate the asylum claims of migrants for Guatemala and make recommendations to the Guatemalan Migration Institute (*Instituto Guatemalteco de Migración* or IGM): a process described in the next section.

But doing so is challenging in a place radically transformed by settler colonial efforts. Guatemala holds its own recent refugee histories, which the Guatemalan government is attempting to navigate in building a national asylum regime. These dynamics are connected to decades of Spanish then US imperialism and political economic interests. Indigenous Q'eqchi' (Guatemala's second-largest indigenous group at almost a million people) are long-standing inhabitants of the Petén, having been migrating there from Alta Verapaz for centuries (Grandia 2012).[13] During Guatemala's 'ten years of spring' (1944–1955), democratically elected President Jacobo Arbenz (1951–1954) initiated a 1952 land reform that gave land to over 100,000 landless peasants and their families (Jonas 2000). Alleging efforts to stop communism's spread, the CIA sponsored a military coup in 1954 to oust the reformist president, and overturn the agrarian land reform. In

---

[13] The Itzá are the original inhabitants of the Petén, who many Q'eqchi' refer to as their 'elder cousins' (Grandia 2012). The Itzá are now few in number and predominately reside in San José on Guatemala's Pacific coast.

actuality, this was done to protect private US business interests, in particular with the United Fruit Company (Colby 2011). Meanwhile, the 36-year civil war (1960–1996) that followed led to the deaths or disappearances of 200,000 people, along with the displacement of over a million more. After experiencing the massacres of Guatemala's civil war, many ethnic Maya (in particular Q'eqchi') fled to the Petén to escape state violence and repeated loss of territory. Many were then driven into refugee situations in Mexico and the US, not returning until the late 1990s (Carr 2008). Now, Q'eqchi'are dealing with repeated attempts to push them from the land. This includes state-led colonisation efforts from the 1960s, driven by coffee plantations and cattle ranches, as well as the creation of national parks that are taking over vast swaths of the Petén (Ybarra 2018). The institutional and structural racism of economic hardship and political and social instability continues to contribute to Maya out-migration (Morris 2022). At the same time, the Guatemalan government is also dealing with record-breaking numbers of its citizens deported back from the US (Golash-Boza and Ceciliano-Novarro 2019). These dynamics overlap in contradictory ways with the environmental changes and imaginaries promoted in Guatemala's asylum frontiers.

The next two sections turn to what Kikon (2019) terms 'living with resource frontiers' when looking at the entangled worlds of oil and coal mining on foothill residents in Northeast India. In the context of Guatemala, *living with* illustrates the conflicting social relations from asylum frontiering efforts. The development of Guatemala's national asylum regime creates tensions related to raw dynamics of refugeeness and return in the region. Excessive American policy interest in Central American migration also allows the Guatemalan government to sideline other types of im/mobilities that are seen as problematic for their image. In particular, this means Mayan civil war refugees and deported Guatemalan migrants, for whom there is little reintegration support.

**Section III: Living with Resource Frontiers in the Petén**
Not long after World Refugee Day, I am sitting in a wooden open-air shelter in the dense tropical rainforest of the Mirador-Río Azul National Park in the Maya Biosphere Reserve: a 2.1-million-hectare reserve that encircles one-third of Guatemala's territory. Several park rangers are lying in hammocks, taking well-earned siestas after a morning spent collecting wood in the thick jungle heat. I am chatting with Miguel, one of the team's newest rangers. Originally from Honduras, Miguel is now working for the Fundaeco team as part of their new *Empleos Verdes* (Green Jobs) programme: a collaboration with UNHCR and the Guatemalan NGO, El Refugio de la Niñez. In the initiative, Guatemalans together with Central American refugees – so far 70 refugees and an equal number of locals – are trained on environmental conservation and ecology and, eventually, employed as forest rangers in ecological reserves in the Petén. The refugee rangers programme – and employment initiatives for migrants and locals at large (particularly young people identified as potential migrants northward) – has received a significant boost of support in recent years from the US government, operating through UNHCR. The previous year, 17% of undocumented border crossers (279,033 people) were recorded from Guatemala: after Mexico and Honduras, the third most significant country of origin in the region (Pew Research Center 2021). Most are leaving decades of criminal violence, but also poverty, natural disasters, and environmental deterioration, contributed to by decades of US imperialism and extractivist practices (Pine 2008). By funding Guatemala's asylum legal system and resettlement opportunities for those who obtain asylum, such as *Empleos Verdes*, the US attempts to encourage migrants to stay in southern regions.

Fundaeco's *Empleos Verdes* is just one of a number of employment initiatives available for migrants (largely from Honduras, Nicaragua, and El Salvador) who have received successful protection claims through Guatemala's newly set up asylum system. In 2023, the number of asylum applications in Guatemala reached a record 1255 since the country established a national asylum system in 2001: but still incredibly low numbers in comparison to other regions (UNHCR 2023). Several of the rangers at the El Mirador camp are refugees whose involvement is funded through UNHCR as a durable solutions strategy.[14] The *Empleos Verdes* program is part of Turi-Integra: a MIRPS collaboration between UNHCR, the Guatemalan Ministry of Labor, and the Guatemalan Institute of Tourism. Local businesses in the tourism sector are encouraged to make jobs specifically available for migrants with Guatemalan refugee status. These efforts are all part of the racialised labour economy of 'putting refugees to work' characteristic of the modes of extractivism rife in the asylum regime (Frydenlund and Cullen Dunn 2022; Martin and Tazzioli 2023). Extracting value from refugees is by no means new. During the post-war period, migrants were accepted by states explicitly as workers under general migration schemes. Now, humanitarian discourses work to contain refugees under the semblance of aid and empowerment. Attempts to provide refugees with rights to work using discourses of self-reliance are part of the shift from humanitarianising refugees to economising them. Refugees are transformed into labour resources, signifying a transitioning of refugee policy from emergency relief aid to economic development programs (Bardelli 2018). As evidence of these steady transformations, several of the restaurants and cafes that ring the island of Flores are partially staffed by refugees from surrounding Central American countries.

*Empleos Verdes* is one such offering designed to provide employment for regional refugees. The programme has gained traction, lauded by Filippo Grandi, the High Commissioner of UNHCR. In his December 2021 visit to Guatemala, Grandi was specially toured through the Tikal archaeological site by a group of refugee rangers. "Guatemala is a country of origin, transit, destination and return. As a transit country, every year thousands of people cross the territory, but more and more people perceive that it can also be a destination for those who have to flee violence and persecution," commented Grandi at the time. Grandi's visit to Tikal inadvertently points to tensions in *Empleos Verdes*. The Petén was long a place of refuge for Indigenous Q'eqchi', who were uprooted and hid in the jungle for years during the decades of civil war violence (Ybarra 2018). However, the creation of protected parks in the 1990s, which *Empleos Verdes* now serves, drove many Maya from the land. That conservation programs have now attracted popular appeal for local refugee integration is a steady pattern of racialised dispossession in the region. Many of the resettlement initiatives available in the Petén are designed to support an elite tourist economy: pushing "the frontiers of commodification" (Devine 2017: 638) into new spaces that reinforce racial and colonial legacies. Regional refugees like Miguel and low-waged Guatemalans (many Indigenous Maya) are tasked with maintaining national parks around the country that are visited by tourist elites. This work is physically and mentally demanding, requiring rangers to be away from their families and friends for days, often weeks, on end. In their efforts to put an end to forest fires and the occupation of park territories, some rangers I met had also experienced death threats from narcos and loggers. To evade the threats of powerful narco ranchers operating in the Maya Biosphere, several rangers obtained asylum from Guatemala to the US. That some rangers have

---

[14] Durable solutions refer to the three so-called lasting solutions of voluntary return, local integration or relocation that drive the international refugee regime. In this case, *Empleos Verdes* is a practice of integration, designed to locally incorporate those who have received successful asylum claims.

obtained asylum *from* Guatemala, while others have received asylum *for* Guatemala is emblematic of the tensions that exist in the country more broadly.

The Guatemalan government has capitalised on US financial support, pushing to make Guatemala a more attractive destination than surrounding regions. The US has become notoriously difficult for making an asylum claim, including a virtual ban at the southern border. Asylum seekers are generally treated as suspect through grueling lengthy interviews (Haas 2023). Asylum claims processing can take several years, often without authorisation to work.[15] Across the border in Mexico, migrants are required to await the results of their cases in the regional state in which they claim asylum. My interlocutors reported that asylum seekers are sometimes deported for leaving that state without authorisation. Because of the overwhelming numbers of cases in southern border regions, such as Chiapas, migrants face challenges in accessing already strained basic services (Vega 2021). Nearby countries such as Belize and Costa Rica have also steadily made claiming asylum a challenging process (Freier and Rodriguez 2021). Asylum seekers face barriers to work and experience mandatory detention, unless they have submitted an asylum application within a short time of arrival.[16] Meanwhile in Guatemala, posters in popular migration routes visibilise the efforts to tempt migrants to claim asylum locally. UNHCR campaigns like 'Guatemala, Opens the Door to A New Beginning' attempt to promote asylum locally.[17] UNHCR mobile unit vans and boats advertising asylum plough across hard-to-access rivers and jungle roads in rural regions. Cultural events such as World Refugee Day have become annually promoted events.

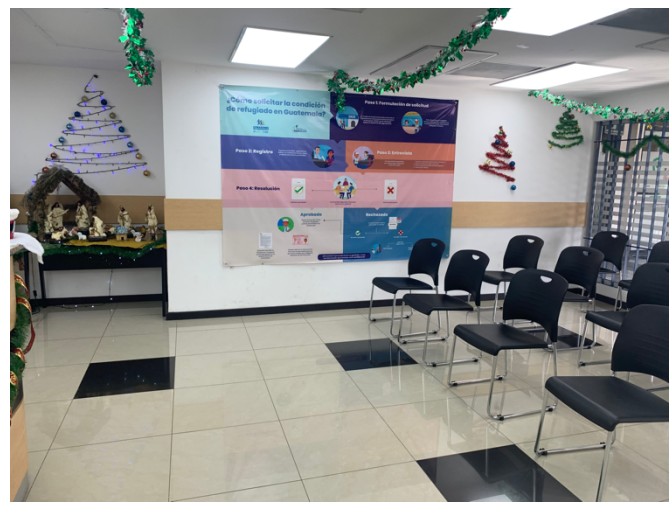

**Figure 4:** *Departamento De Reconocimiento De Estatus De Refugiado* in Guatemala City, photograph by author.

Travelling to Guatemala is easy for nationals of most countries as visas are not required for the majority of foreign citizens. In Guatemala, migrants can then make an application verbally or in writing at any immigration checkpoint, including directly at IGM's Department for the Recognition of Refugee Status (*Departamento De Reconocimiento De Estatus De Refugiado* or DRER) in Guatemala City.[18] This includes describing the situation that forced that individual to leave their home country. Asylum claimants are immediately placed under the status of temporary resident (*Estatus de Permanencia Provisional*), receiving documentation that provides them with access to social services. From the start, they can also work legally while awaiting the results of their applications: an uncommon practice in most Geneva Convention signatory states.

---

[15] In the US, asylum seekers must wait a year after submitting an asylum application before applying for authorisation to work. The wait on employment approval is then an additional lengthy process. See Haas (2023).

[16] In Belize, migrants must apply for asylum within 14 days from the moment of entry; in Costa Rica, the deadline is within one calendar month.

[17] See https://www.acnur.org/guatemala-abre-la-puerta-un-nuevo-comienzo

[18] See IGM's website for a technical description of the asylum process in Guatemala. https://igm.gob.gt/refugio-en-guatemala/

In addition to a small monthly stipend, UNHCR covers asylum seekers' short-term accommodation and provides them with support to find employment. Two weeks after their formal request, applicants travel to Guatemala City for their interviews with the DRER. In what I have described elsewhere as a process of 'intimate extraction' (Morris 2023a), applicants must then detail why their left their country of origin and want to settle in Guatemala.[19] CONARE will later issue a recommendation on that individual's application. The asylum process itself takes a maximum of three months: generally, far less, given the low case numbers being processed by IGM. The success rate is high at 70-75% (compared to 42% in the US), granting refugee status to most applicants. In case of the rejection of a claim, there are also several avenues for appeal: first to IGM and then a judicial route through Guatemala's court system. Guatemala's legal framework also applies to a broader group of people than in other countries (known as complementary protection), including an extension to family members.[20] With a refugee visa, it is alleged that migrants can then easily work, open a bank account, and have access to health and education services. There are also grants available through UNHCR for starting up new business ventures.

But these policy shifts rarely allow for meaningful livelihoods, often just survival. Many of my interlocutors held mixed feelings about these much-promoted opportunities. Some described health care as lacking, especially for psychological and specialised services: essential for those holding harrowing experiences, and something that has also been found as lacking among Guatemalan returnees (Sabin et al. 2006). Others were concerned about educational and future employment opportunities for their children, when poverty is estimated at 55.1 percent of the population (World Bank 2023). Regardless of the fantasy of refugee support, many struggle to find work and secure housing in the long-term. And overall, like Guatemalan citizens, refugees face a stagnant economy with a minimum wage of just over 11 quetzal (US$1.42). It is for many of these reasons why there is little take up of asylum locally and why significant numbers of Guatemala's own citizens continue to leave the country.

For Miguel, part of the appeal of Guatemala's asylum frontier is the challenging transnational labour environment experienced between Central American countries. Tough laws across Central American countries prevent people in low-waged employment from easily working outside their country of citizenship. Under the Central American Agreement for Free Mobility (CA-4), citizens from El Salvador, Guatemala, Honduras, and Nicaragua can move across participating countries but without the right to employment, healthcare, and education. Several migrants I spoke with, like Miguel, while also having sufficient grounds for asylum, did so because they could not easily access a working visa otherwise. Without visa sponsorship from an employer, those in lower-wage or manual sectors (such as agriculture, construction, and service industries) faced long-winded applications for residence and work permits. In this quagmire of bureaucracy, applying for asylum is one – albeit precarious – means of living and working across borders.

Others said to me that Guatemala was not their destination but where they intended to save money before moving further northwards. For these individuals, working in Santa Elena under a temporary residence permit during the processing of their asylum claim can help generate finances for future movement. Yet, because of information sharing provisions across UNHCR and state

---

[19] Ironically, it is Guatemala's and other Central American countries devastating experiences of violence that led to the development of the regional 1984 Cartagena Declaration of Refugees, which forms the basis for Guatemala's national asylum regime.
[20] See Articles 43-45 of Guatemala's *Migration Code*, available at: https://igm.gob.gt/wp-content/uploads/2021/11/1_Codigo-de-Migracion-Decreto-44-2016-del-Congreso-de-la-Republica.pdf

agencies, the possibility of making an asylum claim to the US after having received it for Guatemala is likely limited. Even so, IGM officials told me that many asylum seekers and refugees for Guatemala end up eventually moving elsewhere. Although there are no guarantees, some migrants believe, rightly or not, that having humanitarian legal documentation means not having to evade authorities on the way to the US border. And so, for those without recourse to start-up capital and who prefer to stay in the region, like Miguel, or at least bide their time before attempting to reach the US, receiving asylum in Guatemala can be an appealing option.

Non-governmental and intergovernmental agencies are central to this unique form of frontier capitalism focused on migrants claiming asylum. In the Petén, asylum externalisation is characterised by the involvement of larger refugee industry players such as the UNHCR, the Human Rights Defenders, Red Cross, and Doctors Without Borders. These familiar faces, who I encountered in other global regions as part of the mobile labour of the asylum industry (Morris 2023a), combine with local NGOs and government agencies new to refugee work, such as El Refugio de la Niñez, Conamigua, and IGM. Through their industry engagement, these organisations contribute to the institutional development of asylum. This entails the steady proliferation of Migrants and Refugees Assistance Centers (known as CAPMiRs) across Guatemala through to the refugee-focused projects that I observed in the Petén. For UNHCR, there are pressing financial motivations to consider. In 2023, 96% of UNHCR Guatemala's $22.8 million government funding came from the US government. UNHCR staff I spoke with expressed these tensions in conversation. Some felt that an asylum system in Guatemala could be helpful in particular cases, such as gang and gender-based violence or police abuse. However, others described these forms of brutality as easily following migrants across borders. Gangs span state territories (including many originating in the US), they continued, and Guatemala itself has systemic problems with gender-based violence.[21]

Nor is Guatemala's economic and demographic circumstances positioned to accept large numbers of refugees. One of my UNHCR interlocutors slyly remarked that agreeing to US directives enabled them to fund other "more beneficial" projects locally, such as a childcare center in Santa Elena for local and refugee families. Organisationally, this representative continued, it is in their interests to enhance Guatemala's national asylum regime, rather than labour migration pathways more generally. Refugees are the major source of revenue for UNHCR and other refugee sector agencies. Branching out beyond their market niche could dilute their mandate. But, as the next section shows, this refugees as resource focus has generated tensions locally related to Guatemala's raw dynamics of refugeeness and return.

**Part IV: Frontier Frictions in the Petén**
Not long after World Refugee Day, I walk through the fruit and vegetable market in the main town of Santa Elena, just across the causeway from Flores. Tropical fruits like papayas, rambutans, and zapotes are piled high on ramshackle market stalls. The slap of corn tortillas shaped between street vendors' palms beckons the lunchtime crowds for freshly made quesadillas. As I select papayas for the weekend, I strike up a conversation with the stall's vendor, Estella, about my research. Estella has much to say on the asylum campaigns that are visibly evident across the town. Like many of the market workers, Estella is Maya Q'eqchi'. Like so many middle to older aged Maya,

---

[21] See Bruenau et al. (2011) on the development of *mara* street gangs, which originated in California in the 1980s and were exported to Central America in the wake of US deportations during the 1990s.

she also spent years in exile as a refugee. Unprompted, Estella animatedly describes her memories of being a young girl when the civil war broke out. Estella and her family first settled in the Petén to escape state violence and repeated loss of territory. But revolutionary guerilla forces also used the Petén's northern rainforests to hide from the Guatemalan army. Like other parts of Guatemala, they were drawn into the conflict against their will. The 'scorched earth campaign,' advanced by General Rios Montt, led to over 200,000 deaths, while close to a million people were displaced (CEH 1999). Widespread village-level massacres occurred in overwhelmingly Mayan regions – over 80% of those killed were Indigenous Maya.

Caught in the gunfire between guerrilla forces, Estella and her family eventually crossed the border to Mexico in the 1980s in hopes of survival. "Most of my life, I was in fear of the military finding my family," she says. "We moved between refugee camps in Mexico, almost every year it felt like. Chiapas, Campeche, Quintana Roo," she slowly counts the different states on her fingers. "We never knew if the Mexican government was on our side or if we might be handed over to the Guatemalan army. The Guatemalan army was always coming into Mexico, looking for guerrilla soldiers in the camps. We knew that if they found us, we would be killed without any mercy, we'd seen this happen to others." After nearly a decade living in different refugee camps in southern Mexico, she eventually returned to the Petén as a middle-aged woman in 1995, not long after the peace process negotiations.

All those years spent living in exile in refugee camps before returning the Petén, gives Estella pause when discussing the new refugee programs. "It confuses me. These people aren't refugees. They haven't been through the kind of suffering we have," she says. "Many Maya never got their land back and still live as refugees in Guatemala. They have all these programs for the new refugees, and make such a big deal about them, I've heard they get a month of accommodation, but we Maya experience so much inequality that has never been resolved." The concerns that Estella expresses are unsurprising: Guatemala still has one of the most unequal systems of land tenure in the world. Maya like Estella persevere every day, with recent genocidal histories still fresh in their mind. Now, many face forms of economic exclusion perpetuated by structural racism (Ajcalón Choy et al. 2020). Palm oil plantations, taking over great swaths of northeastern Guatemala that are home to Q'eqchi' communities, are continuing to fuel displacement. So too, environmental changes are impacting on local agricultural yields, pushing some farmers to forsake their lands to find livelihoods opportunities elsewhere. For some, this means making the long journey northward through Mexico to the US.

To this day, Guatemala has the second highest number of people deported from the US after Mexico. More than 300,000 migrants were returned from Mexico and the US back to Guatemala between 2019 and 2022 (IOM 2023b). During my fieldwork, these numbers increased, as the Mexican government was surreptitiously bussing migrants from Guatemala and Honduras to the Guatemalan border of El Ceibo. I met several migrants who had been subject to those covert forced deportations. Indeed, at large, most Guatemalans I spoke with had a migration story to tell in the face of difficult local conditions: of dangerous journeys spent hiding in trucks underneath fruits, vegetables, and other produce; of the going rate to make it through Mexico and across the US border. Many of these residents struggle to provide for their families, having experienced poverty, in some cases violence, and in the past armed conflict. With these overlapping experiences of refugeeness, economic instability, and return locally, it is unsurprising that tensions exist around

the Petén's asylum frontier, such as those voiced by Estella. It is for these very reasons that the Guatemalan government treads gently around publicly calling resettled regional migrants 'refugees.' Some officials I spoke with at Guatemala City's IGM emphasised avoiding the term 'refugees.' Not only does it evoke traumatic memories for many, but the government does not want public perceptions that foreign refugees are preferentially treated.

The development of Guatemala's asylum regime also clashes with the inability of many Guatemalans to access meaningful livelihood opportunities locally. Returned Guatemalan migrants face little reintegration support when back in the country, such as employment assistance (Roldán Andrade 2014). Many struggle to find jobs upon return and apply the skills they have acquired while abroad. On one occasion, I met an older Guatemalan in Flores, who had been deported from the Mexico-US border. He was left without access to food or shelter, with the local *Centro de Migrantes* only supporting foreign migrants, and mostly Hondurans. One local Guatemalan government official recognised these concerns, continuing, "It is a balance of not angering local populations. Making sure to develop programs that also benefit locals. So that the local community sees refugees as drivers of development." A similar narrative was also echoed by UNHCR representatives I spoke with in the Petén: part of a trend towards 'refugee entrepreneurism,' where refugees are explicitly marketed as a 'resource' (Easton-Calabria and Omata 2018). But the emphasis on refugees as harbingers of economic prosperity, either through their labour power or through the financial support their presence engenders, glosses over the structural causes that accounts for people's displacement. As Nora Bardelli (2018) argues, the focus on labour and capital investment frames the 'solution' to displacement in developmental and market-based terms. Displacement has become "a matter of access to the job market rather than a political question about inequalities, exclusion, conflict, exploitation, [and] asymmetrical power relations" (Bardelli 2018: 55).

Yet for the Guatemalan government, foreign asylum seekers and refugees hold political economic value. Guatemalan politicians strategically utilise this migration governance arrangement to offset stereotypes of dangerous conditions locally, attract humanitarian investment, and US development support. Guatemala still retains an image of civil war and present-day violence. Asserting themselves as a place of refugee resettlement is an attempt to cultivate their legitimacy and image at home and abroad. In the Petén, it is also a way for the local government to affirm their sovereignty in a context of historical tensions between bordering Belize and Mexico.[22] Profit is made from human bodies – both by encouraging regional migrants to claim asylum locally and through the local labour that refugees provide. This is a different form of what Kaushik Sunder Rajan et al. (2012) call the 'capitalisation of life.' The Guatemalan government profits off the presence of refugees within its borders, wherein migrants are given financial value in their legal classification as refugees. Migrants also contribute to flows of money and systems of value through their labour power and engagement in the local economy. As one IGM representative pointed out to me in conversation, the Guatemalan government has a long history of strategically engaging with the US on migration. More refugees in Guatemala means more American financial investment, as does consenting to increased border enforcement locally. But this support is tinged with power imbalances symptomatic of a longstanding neocolonial relationship. The US

---

[22] The Guatemalan and Mexican governments have long disputed the location of their border, in particular in the northeastern Petén (Devine 2018). Guatemala and Belize's border disputes also date back centuries. Although Belize achieved full independence from Great Britain in 1981, Guatemala did not recognise Belize's independence until 1991. Subsequent Guatemalan administrations still push to claim half of Belize's territory.

government has threatened trade sanctions and the withdrawal of aid to Mexico and Central America, and even shutting down the border entirely, if these countries do not enhance their border security measures. This has placed local leaders in a tricky position of balancing humanitarian and logistical priorities and political pressure from US officials keen to extract political value from the prevention of undocumented migrants.

Despite these pressures, asylum frontiers also come with immense *moral* value. For several IGM representatives I spoke with, moral considerations play a substantial role in their desire to support the development of a local asylum regime. One described running clothes and food drives for asylum seeker families in Guatemala City. They emphasised the moral value of this regime in supporting those in need from surrounding regions. These moral values embedded in the international refugee system speak to the 'resource politics' rendered visible in asylum frontiers: a term commonly used by political ecologists to describe contestations over access, use and control of resources (Watts and Peluso 2013). Although driven with moral conviction, this narrative of welcome – what Krista Johnston (2022) refers to as 'settler care' when examining the arrival of Syrian refugees to Winnipeg, Canada vis-à-vis long-standing Indigenous communities – sits in uneasy contrast to the ongoing exclusions experienced by Maya across the country. By prioritising refugees, the Guatemalan government can gloss over the immense socio-economic disparities experienced by many Indigenous Maya, who are not afforded dominance in the settler colonial order of Guatemala's *ladino*-majority elite.[23] Meanwhile, the neocolonial project of Guatemala's asylum frontier decontextualises the histories of US imperialism in the region that have induced precarity and driven people to migrate.

These resource politics also extend to larger contestations over access to and control over refugees as resources. As I found in Nauru, many western solidarity activists and media spotlight Guatemala as a site of extreme human rights abuse, "not safe for refugees … where thousands of people are desperately … fleeing violence and persecution … [and] face extremely high rates of murder" (Human Rights First 2019). Such humanitarian narratives valorise the refugee regime as a racialised regime of white care (Picozza 2022), where advocates vie over global access to, control over and use of *refugees*. In these struggles for resource control and ownership, largely white liberal westerners – not people of colour or Global South nations – are represented as the rightful rescuers of Black and Brown refugees. The figure of the refugee becomes a subject of contestation because of its value-generating potential. Ironically, such moralising and racialised representations only boost the values that contributes to the development of asylum frontiers (Morris 2019). When I asked IGM representatives why enhance Guatemala's asylum regime and not labour migration more generally, they pointed to the morality of developing such an arrangement regionally. Asylum, they continued, also serves a governance function in that it allows the Guatemalan government to better control incoming migrant numbers. Individuals are scrutinised through humanitarian classifications of vulnerability to identify who is most in need. This veritable system of resource governance is easily advanced because of the institutionalised fabric of the refugee industry. Established refugee industry knowledge making and mobile labour forces of personnel advances asylum into new frontiers.

---

[23] In the settler colonial state of Guatemala, national progress remains bound up with white sociospatial epistemologies (Loperena 2017). Such ideologies have long been used to negate Indigenous territorial claims and buttress the political and economic aspirations of the *ladino* elite.

Such resource politics provoke important normative considerations. Protection pathways such as asylum and resettlement are so often insufficient for the compounding factors that induce people to migrate, as is the case with many people on the move in Guatemala. Many are ineligible for the strict criteria of these legal pathways and do not neatly fit in the corners of an asylum claim. So too, asylum is now all-too-often undermined by states brokering deals into new frontiers. Regardless, what we see is that the asylum regime in Guatemala is of little appeal, despite all the money and resources invested. Ultimately, Guatemala's asylum system fails to address how and why people have been rendered into refugees in the first place. But even as asylum dehistoricises such factors, generating multiple levels of injury in the process (Haas 2023), the moral power of the figure of the refugee rarely allows for these kinds of conversations. Unravelling this emotional investment may be the principal challenge for advocating for other forms of safe passage.

There are powerful economic incentives for both the Guatemalan and US governments to facilitate legal migration pathways. In 2023, remittances to Guatemala totaled a record $20 billion, mainly from the US, which comprises almost 20% of the country's entire economy (World Bank 2023). For the American government, Guatemalan labour and skills – often unfairly low waged – are also indispensable to the US and global economy (Frydenlund and Cullen Dunn 2022). Nonetheless, despite the demand for jobs abroad, opportunities for many Guatemalans (especially Indigenous Maya) to move and earn money abroad through legal pathways are limited. The emphasis on preventing irregular migration yet promoting asylum protection detracts from these lived realities, where many people see irregular migration as less dangerous than the status quo. And indeed, underneath their surface differences, these practices of border enforcement are shaped by a common set of forces grounded in what Mezzadra and Neilson (2019) term the 'operations of capital': a massive global assemblage of extraction, finance, logistics, state and non-state power. Refugee industry agencies continuously invest in asylum expansion, which engenders types of spaces such as in the Petén – that of asylum frontiers. The development of specialist mobile workforces trained in refugee status determination and resettlement through to cultural propaganda on the benevolence of asylum enables the movement of resource production assemblages across frontiers. These extractive frontiers quite often proceed apace, ignoring previous residents, in ways that correlate strongly with processes of wild extraction and destruction characteristic of resource frontiers at large.

**Conclusion**

Political ecology has long examined questions of natural resource extraction and the rapid transformation of remotes spaces into often socially and ecologically destructive production sites. Forest and agrarian spaces are imagined as places with vacant lands open for settlement and abundant natural resources. More recently, migration studies has paid attention to accumulation practices grounded on human (im)mobility, which can also have devastating consequences. By bringing these insights together, this article demonstrates how political economy is crucial for understanding the formation of southern migration policies. The production of refugees occurs because of the ravages of global capital, where people struggle to survive and become enveloped in the prospecting logics of converging humanitarian and migration governance regimes. Today's migrations result, in part, from US imperialist machinations of extractive practices. Situating refugees in global capitalism urges on considerations around how to break up the next wave of extractive asylum economies.

The resources that drove Guatemalan and imperial interests in the Petén – coffee, palm oil production, oil extraction, and biodiversity – revolve around questions of frontier-making. Here, I have shown how frontier practices of accumulation centre on the bodies of migrants too, in their classification as refugees. Migrants are labelled and disciplined as refugees under material regimes of practice and control. States leverage their position as host states of displaced communities, extracting revenue from other states for maintaining migrant groups within their borders on the basis of ascribing prices to refugee bodies. However, the effectiveness of this is not always realised in practice. Migrants potentialise the logic of commodity in their self-appreciation as capital in ways that are related to their labour power and fulfilling their onwards goals. This system of unfree labour occurs in a context where racial, classed, gendered, and ethnic biases structure how people can move across borders and their reception in what Aihwa Ong (1999) terms 'graduated sovereignty.'

The Petén's new kind of frontier-making is entangled within a massive global assemblage comprised of diverse actors and agents. But rather than painting a homogenising picture of border externalisation policies, this article foregrounds the political and social heterogeneity of this geographical area. The asylum frontiers that this article details in the Petén rub up against continued dynamics of local displacement and return. Frontier extraction, political ecologists show, often creates fractured ecologies. These state-sanctioned practices advance without regard to Indigenous peoples or migrants' aspirations – making the extractive frontiers here comparable to Tsing's extractive Bornean frontier. What we need to do in migration studies is consider more critically the economies and ecologies that take shape when migrants are the resource. It is not only non-humans that result in the reworking of frontier regions but also mobile human life in relation to demands for economic development. We are grappling with very different frontier moments, where western governments and southern elites are pushing border-making into new contexts. Novel socio-economic practices are generated in often remote areas as asylum systems are cropping up where once other forms of colonial extraction dominated (Morris 2021b). Understanding these spaces as resource frontiers spurs a mode of engagement that makes visible how asylum constitutes a frontier of market capitalism.

Concerns from refugee solidarity advocates surrounding the erosion or devolution of western asylum belie how this hyper-extractive system is a method of border control. The ways that asylum sits alongside other practices of migration deterrence in the Petén compels such understandings. The capitalist appropriation of refugee bodies is also contingent on mobilising imaginaries of white western salvationalism. These ideological drivers of asylum marketisation contribute to the material generation of new asylum frontiers. In many respects, the Petén is still a wildcat frontier characterised by the dispossession of marginalised subjects. But frontiers can throw up alternatives, suggesting the importance of thinking beyond asylum and its material and discursive dimensions. What free movement regimes might enable a more just mobile future? Towards that end, chronicling the frictions produced in emergent asylum frontiers, as well as why migrants and residents are engaged in these resource-making projects, offers transformative possibilities for moving beyond asylum.

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
