# Peer review of "“Reborn in Guate”: Making Resource Frontiers in Asylum in Guatemala’s Northern Petén"

_Migration Politics_

## Round 1 · Referee Report · Anne McNevin (Referee 1) · 2024-4-14

Report

This is really an excellent article and I have very little criticism to offer. It was very thought provoking for me and I think it’s largely publishable as is. The article makes a convincing case, in my view, for the analytical value of an “asylum frontier.” The article is engaging, well-written and structured; it is well introduced, including methodologically, and represents an impressive crystallization of Morris’s extensive fieldwork over a number of years. The case of Guatamala and the Peten are well-situated within a broader global context, making both the general argument more compelling and significant, but without losing – and indeed emphasizing – the specific contradictions and tensions operative amongst the particular asylum frontier in question. In my view, the article succeeds in making the three advances at the intersection of externalization and migration-industry literatures: (1) attention to an assemblage of ‘drivers’ rather than isolated actors; (2) focus on value rather than profit alone (I especially appreciate this feature, since it expands out a political-economy reading to incorporate moral and symbolic economies); and (3) the reproduction of racial and colonial hierarchies via critique itself.

I’m left with two questions that might offer some scope to refine the general argument – though as I note, it’s already very effectively put. The first is about the notion of “putting refugees to work” (p.10) which is certainly a feature of recent strategies of migration management in Jordan and elsewhere, as Morris’ other work has shown. I’m wondering about an earlier phase of migration governance , prior to the inauguration of a separate “humanitarian” stream of migration in which post-war migrants and refugees were accepted by states explicitly as workers under general migration streams. On one hand this means that getting value from refugees is by no means a new thing – but there’s certainly enough that’s particular to the way it’s done today to make the ‘asylum frontier’ meaningful in terms of its relation to capital. Is the suggestion though, that the continuities here are somehow covered over by the mask of humanitarianism? Does that make it more pernicious somehow?

I think what I’m getting at here is part of a second and related question, and that’s the extent to which the argument as a whole is an analytical and/or normative one. The analytical dimension is clear (and to me the most compelling) and spelt out in the conclusion. By investigating asylum as resource frontier, scholars become more attuned to the extraction of value as a driving force for offering, seeking and regulating asylum and we can become far more critically attuned to humanitarian justifications as well as performances of refugeeness. (I especially like the way Morris implicates everyone in value extraction, including migrants.) But whereas Morris offers no judgement of the latter – which may be entirely appropriate – there is an implicit judgement of the regulators that’s never fully spelt out. There’s a hint on p.16, that it would somehow be less duplicitous (?) to just have general migration schemes rather than this charade of asylum (?) – a scenario that might return us to the post-war examples mentioned above. Or is the key critique made in relation to northern states who use new asylum frontiers to effectively contain migrants? Is there an implicit case being made for open borders? I don’t think you have to necessarily develop such a case in any detail. It’s just that I was not ultimately sure whether this was intended to be primarily an analytical contribution – and on that front I think it’s really effective – or whether there was more going on in terms of normative critique. If the latter – then the assemblage argument potentially hinders a clear view of where the power and responsibility lies – or at least makes it necessary to clarify; and I think one or two well placed sentences to state your position early on would sharpen and enhance the article.

Thanks for the chance to read this and looking forward to seeing the final version in print.

Recommendation

Publish (surpasses expectations and criteria for this Journal; among top 10%)

---

## Round 1 · Referee Report · Anonymous (Referee 2) · 2024-4-20

Strengths

  1. This article presents a fascinating topic such as externalisation/new precarious extractivist asylum regimes with an understudied case like Guatemala.
  2. Its rich theoretical approach allows for a critical reflection on the intertwining of mobility restrictions, the punitive migration system, and the externalisation of asylum in conditions of risk that favour the extractivism of governmental and non-governmental actors, especially international agencies and local governments, through the mechanisation of asylum.
  3. Very rich ethnographic material.
  4. Your arguments for considering the externalisation of asylum as resource frontiers are compelling.

Weaknesses

  1. The main argument can be improved by better defining and prioritising some concepts so that the analytical focus of the text is on asylum frontiers in the context of the functioning of resource frontiers.
  2. It can be confusing to talk about new asylum regimes; someone unfamiliar with the legal figures of international refugee law needs more context as to what asylum versus complementary protection entails. What does this tension between facilitating regular channels for migration and emphasising asylum as the sole regularisation measure imply? Why has asylum become a migration regularisation measure?
  3. I don't think the argument about how migrants use state framings to advance their frontiers is successful. It seems to me that the paper concentrates on extractivism. Although it does problematise migrants' decisions, for which Miguel's case is extraordinary, it seems to me that it is for another paper to discuss strategies of resistance to this imposed system of forced asylum in the asylum industry in Guatemala.

Report

I really liked this article, I learned a lot. I think it is very important that you have addressed the case of Guatemala. Your analytical framework is very rich and your contribution is excellent from a perspective of the externalisation and weakening of asylum under the fiction of an asylum system at the resource frontiers that allows for the abundance of an extractivist industry of forced migrants. I suggest some changes for clarification, especially if you will be read by people working in the area of asylum law and people unfamiliar with the Central American region.

Requested changes

  1. It is necessary to better explain the context of mobility of the people on the move in Guatemala, and how the context of humanitarian aid and international protection forces them to modify their narratives. It must be said that UNHCR requires narratives on violence and that the causes of migration are multiple: it is criminal violence, but also poverty, natural disasters, environmental deterioration, and as these are not causes in the Geneva framework, they cannot be in their narratives.
  2. I wonder if Guatemala uses the Cartagena Convention to process applications; it should be clarified in the text.
  3. It would be beneficial to compare the numbers of asylum seekers in Mexico because that will allow you to demonstrate more forcefully that people are not staying in Guatemala. You mention that the numbers are low, but if we compare it with 141,053 asylum applications in Mexico 2023 and only in January 2023 a total of 13,113, what we see is that the programme in Guatemala does not really work regardless of all the money and resources invested and produced by the "asylum frontier"; of course, it does not have the capacity to receive the numbers that Mexico gets. But it seems it will be useful to mention because most people applying for asylum in Mexico pass through the Petén area.
  4. It is necessary to explain in more depth how the US is violating its commitments to international refugee law under the guise of legality with its asylum outsourcing practices. This is important because, in the conclusions, you write that Guatemala, as a host country, benefits from this programme, but it seems to me that the US benefits the most. After all, it simulates a policy of support, knowing that the programme will not achieve its objectives because the country does not have the possibilities or conditions to be a safe destination. In the end, the filter that follows is detention and deportation in Mexico.
  5. Check the term asylum frontier, because sometimes you use the term asylum industry frontier.
  6. I don't know if the UNHCR programme is part of the durable solutions framework, but I imagine it is. It needs to be explained because applying for asylum and the real experience of integration can be confusing for someone unfamiliar with Central America's context. Specially in this case, we want to know more about the tensions produced in daily life with returned/deported migrants and internal displaced people.
  7. I understand that you are talking about both the externalisation of asylum and migration. But it seems to me that you should address directly the externalisation of asylum, otherwise can be confusing. However, when you argue that these people are not necessarily refugees, you should explain why you use both terms, how they are related, why the boundaries between forced migration and asylum are no longer clear, and what's more, they hinder the reality of having very precarious mechanisms or intl protection and almost none for regularisation.
  8. At several points in the text, you mention that asylum is reinforced and promoted, I believe you must be more critical. Is asylum being promoted, or is the industry and the economy around asylum? In that case, they are not promoting comprehensive, dignified protection within a social justice framework, so my advice is to be more critical and say that they are promoting a precarious figure of protection, more similar to migratory regularisation, without wanting to call it that.
  9. I suggest taking the paragraph on page 5 (your work in Jordan and Nauru ) at the beginning of the paper to understand how and why you came to Guatemala.
  10. Explain better, with some examples, the processes of exploitation and administrative violence baked into the racialised mobility regime of the international refugee system.
  11. What do you mean by "embodying resource frontiers as a subversion of asylum controls?
  12. I don't think it is the perpetual expansion of asylum borders; I think it is the perpetual expansion of a very precarious form of international protection, which weakens the notion of rights and is a mix between temporary humanitarian aid and a form of migration regularisation that allows the exploitation of racialised migrants.
  13. I also think it should be mentioned that many of the migrants have also left their countries because of extractivist practices; the case of Honduras is obvious.
  14. The term humanitarian migrants needs to be explained.
  15. When you mention the case of Mexico, the legislation establishes that the person cannot leave the state where the procedure begins but not the city. In practice, they cannot go to another city either because the procedures are centralised and costly, but the law only establishes that the person can not leave the state.
  16. It needs to be clarified in which country it takes 14 days to apply for asylum from the moment of entry; if this is the case in Mexico, the deadline is 30 days.
  17. Explain what other social services they have access to. The law says so, but can they access medical services, education, etc. in practice?
  18. I suggest to explain from the beggining that one of the great contradictions of this new asylum system or this asylum frontier is that there are no policies or programmes for returnees or internally displaced persons.

Recommendation

Ask for minor revision

---

## Round 2 · Author Response

Thank you to the reviewers and editor for their supportive feedback. I greatly appreciate everyone's time and effort in supporting this paper’s development. It is greatly improved as a result.

---

## Round 2 · List of Changes

I have addressed both reviewers’ suggestions as follows:

Reviewer 1: 1. I have better detailed the earlier phase of migration governance when value was also extracted from refugees under general migration schemes. This includes distinguishing the new phase of value extraction/asylum frontiers as one connected to humanitarian sentiments (pg. 12). 2. As suggested, I have clarified the analytical and normative dimensions of my argument by explicitly stating it in the introduction (pg. 3), towards the end of Section IV (pg. 19), and the end of the conclusion (pg. 20), as well as being more critical of northern states’ use of asylum (pgs. 8, 17-18) and the inability for the arrangement in Guatemala to work (pgs. 14-15).

Reviewer 2: 1. I have better explained the context of people on the move in Guatemala, including the dictates of the Geneva Convention (pgs. 6 and 18). 2. Guatemala’s use of the Cartagena Convention is clarified in note 19 on pg. 14. 3. I discuss the comparison between the numbers of asylum seekers in Mexico and Guatemala to better strengthen my argument and show how most migrants are not staying in Guatemala (pgs. 9, 16, and 19). 4. I better detail the US’ violation to it commitments to international refugee law (pg. 8) and clarify the disproportionate benefit to the US (pgs. 17/18). 5. I have removed the term ‘asylum industry frontier’ to avoid confusion. 6. I have better explained the notion of durable solutions (note 14 on pg. 12). 7. I have removed the term ‘migration externalisation’ to avoid confusion. 8. By stating my argument from the outset (pg. 3), I have clarified that it is asylum being promoted by industry actors owing to the value economy around it. This includes noting that it is a precarious system of protection that is being promoted. 9. As well suggested, I have reorganised the introduction to move my past work in Jordan and Nauru to the beginning of the paper. To ensure that the introduction remains succinct, I have shifted paragraphs around my theoretical contributions to section 1. 10. I have clarified the processes of exploitation and administrative violence baked into the racialised mobility regime of the international refugee system (pg. 5). 11. I removed the phrase “embodying resource frontiers as a subversion of asylum controls” and the resistance aspect of the paper. 12. By clarifying the paper’s argument (pg. 3), I have addressed the point around the precarity of the expansion of asylum. 13. On pgs. 6 and 11 I mentioned that many migrants have also left their countries because of extractivist practices. 14. I removed the term ‘humanitarian migrants’ to avoid confusion and excessive definitions. 15. I clarified the case of Mexico and that it is the state that asylum seekers cannot leave (pg. 13). 16. I have clarified that it is Belize where migrants must apply for asylum within 14 days from the moment of entry (pg. 13). 17. I have better explained what social services refugees have access to in practice on pg. 14. 18. I have better explained from the start that a major contradiction of Guatemala’s asylum system is that there are no policies or programmes for returnees or internally displaced persons (pgs. 3 and 5).

---

## Editorial Decision

unknown